# Visible Light Driven Heterojunction Photocatalyst of CuO–Cu_2_O Thin Films for Photocatalytic Degradation of Organic Pollutants

**DOI:** 10.3390/nano9071011

**Published:** 2019-07-13

**Authors:** Negar Dasineh Khiavi, Reza Katal, Saeideh Kholghi Eshkalak, Saeid Masudy-Panah, Seeram Ramakrishna, Hu Jiangyong

**Affiliations:** 1Faculty of Biosciences & Medical Engineering, University Technology Malaysia, Johor 81310, Malaysia; 2Department of Civil & Environmental Engineering, National University of Singapore, Singapore 119260, Singapore; 3Department of Mechanical Engineering, Center for Nanofibers and Nanotechnology, National University of Singapore, Singapore 117575, Singapore; 4Electrical and Computer Engineering, National University of Singapore, Singapore 119260, Singapore

**Keywords:** photocatalytic degradation, CuO, Cu_2_O, heterojunction

## Abstract

A high recombination rate and low charge collection are the main limiting factors of copper oxides (cupric and cuprous oxide) for the photocatalytic degradation of organic pollutants. In this paper, a high performance copper oxide photocatalyst was developed by integrating cupric oxide (CuO) and cuprous oxide (Cu_2_O) thin films, which showed superior performance for the photocatalytic degradation of methylene blue (MB) compared to the control CuO and Cu_2_O photocatalyst. Our results show that a heterojunction photocatalyst of CuO–Cu_2_O thin films could significantly increase the charge collection, reduce the recombination rate, and improve the photocatalytic activity.

## 1. Introduction

Rapidly increasing concentrations of organic pollutants such as pesticides, personal care products, and pharmaceuticals into different kinds of water sources have widely increased concerns over the potential impacts of these pollutants [1,2,3]. Conventional wastewater and water treatment cannot effectively remove organic pollutants due to their low biodegradability [4,5]. As a result, these pollutants are considered a serious threat to aquatic ecosystems and public health [6,7,8,9].

Different methods such as the heterogeneous photocatalytic process [10,11], heterogeneous catalytic oxidation with H_2_O_2_ [12], Fenton/photo-Fenton oxidation [13,14], ozonation [15], and UV/H_2_O_2_ treatment [16,17] can be used to destroy organic pollutants. Among these methods, the heterogeneous photocatalytic process is the most promising technique to remove organic pollutants from aqueous solutions because of its reusability, long term applicability, and low maintenance [18,19]. Due to the excellent chemical and physical characteristics of transition metal oxides, these materials are promising candidates in fabricating photocatalysts [20,21,22,23,24,25,26]. 

Both CuO and Cu_2_O are considered as the most attractive photocatalysts for the photodegradation of organic pollutants due to their low fabrication cost, high optical absorption, and optimal optical bandgap for visible driven photocatalytic activity [27,28,29,30]. CuO and Cu_2_O are the two main types of copper oxide. CuO is an indirect bandgap semiconductor with an optical absorption depth of around 500 nm, an optical bandgap of 1.7 eV, and a carrier diffusion length of around 200 nm, while Cu_2_O is a direct bandgap semiconductor with an optical absorption depth of around 1000 nm, an optical bandgap of 2.1 eV, and a carrier diffusion length of around 500 nm [31,32,33,34,35]. 

Various methods such as spin coating, electrochemical deposition, deep coating, electron beam evaporation, and sputter deposition can be used to prepare the thin film of CuO and Cu_2_O. Among the different deposition methods of copper oxide, sputter deposition has the highest industrial compatibility, scalability, and repeatability [36,37,38,39,40,41].

Several studies have reported the application of CuO and Cu_2_O for the photocatalytic degradation of organic pollutants in aqueous media. The CuO/zeoliteX was used for the photocatalytic degradation of methylene blue and o-phenylenediamine under sunlight irradiation [42,43]. CuO/SiO_2_ was successfully used for MB degradation in a system under UV irradiation and in the presence of hydrogen peroxide [44]. In one study, the synthesis of the Cu_2_O/carbon nanotubes was reported, which indicated considerable photocatalytic activity in phenol degradation in an aqueous solution [45]. Katal et al. reported the preparation of a high-efficient thin CuO film by in situ thermal treatment and nanocrystal engineering that indicated superior photocatalytic activity for MB degradation under visible light irradiation [23]. The considerable reusability and stability of this photocatalyst provide a new option for thin film based photocatalysts for industrial applications [23]. In another study, the fabrication of nanocrystalline CuO thin films was performed using RF magnetron sputtering; based on some special properties of this photocatalyst including surface defects and oxygen vacancy, the CuO thin film indicated remarkable photocatalytic performance for MB degradation [46]. 

The significant difference between carrier diffusion length and optical absorption depth is one of the main reasons for the high recombination rate and low charge collection of both the CuO and Cu_2_O photocatalysts. Therefore, it is of great interest to develop a copper oxide photocatalyst in a way that reduces the recombination rate and increases the charge collection efficiency. In this paper, by integrating Cu_2_O and CuO thin films, we prepared a heterojunction photocatalyst CuO–Cu_2_O thin film with a low recombination rate and an enhanced charge collection to improve the photocatalytic activity of copper oxides.

## 2. Materials and Methods

All instruments (Singapore) used in this study for sample characterization were the same as those in our pervious publication [23]. Stoichiometric CuO and Cu_2_O targets were used to deposit thin films of Cu_2_O and CuO. All deposition was performed at a radio frequency sputtering power of 200 W and working pressure of 3 mTorr. Glass and fluorine doped tin oxide (FTO) coated glass substrates were used to prepare the samples. 

Incident photon-to-current efficiency (IPCE) was carried out under standard light illumination of 300 W from a xenon lamp with an integrated parabolic reflector.

The photocatalytic degradation process is as completely described in our previous study [23]. A 300 W xenon lamp with a cut-off filter was used as the visible light source. 

## 3. Results and Discussion

The cross-sectional transmission electron microscopy (CS-TEM) image of the fabricated CuO–Cu_2_O photocatalyst is presented in Figure 1a. The overall thickness of the copper oxide photocatalyst was around 500 nm, which is similar to the carrier diffusion length of Cu_2_O and the optical absorption depth of CuO. In addition, the thickness of the CuO thin layer was around 200 nm, which is similar to the carrier diffusion length of CuO.

Due to the lower optical absorption of Cu_2_O than CuO, a greater bandgap of Cu_2_O (around 2.2 eV) than CuO (around 1.6 eV), and a longer carrier diffusion length of Cu_2_O (around 500 nm) than CuO (around 200 nm), Cu_2_O was deposited on top of the CuO thin films (Figure 1b). Indeed if CuO is deposited on Cu_2_O, the majority of carriers may recombine and cannot reach the conducting electrode, which results in the reduction in the performance of the heterojunction CuO–Cu_2_O photocatalyst.

X-ray photoelectron spectroscopy (XPS) measurements were performed on top of each layer to verify the deposition of the CuO and Cu_2_O thin film. Figure 2 shows the Cu2p XPS spectra of the bottom layer of CuO and the top layer of Cu_2_O. Main XPS peaks at 933.9 eV were ascribed to the 2p_3/2_ peak of Cu^2+^, indicating the deposition of CuO [47]. Shoulder peaks at higher binding energies further confirmed the deposition of CuO, while the main XPS peak at 932.5 eV corresponded to the 2p_3/2_ peak of Cu^+^, confirming the deposition of the Cu_2_O thin film [48,49].

The XRD spectra of the prepared CuO–Cu_2_O sample was measured to further investigate the material properties of the prepared photocatalyst. Figure 3 shows the XRD spectra of the CuO–Cu_2_O thin films and the 500 nm CuO and Cu_2_O control thin films. Main XRD peaks at 36.45 and 38.76 degrees were ascribed to Cu_2_O(111) and CuO(111), respectively [5-667 and 5-661 JCPDS-ICDD]. As can be seen, in the CuO–Cu_2_O sample, both XRD peaks of Cu_2_O(111) and CuO(111) can be observed, indicating the existence of both kinds of copper oxides.

To investigate the impact of the integration of CuO and Cu_2_O on the optical properties of the fabricated photocatalyst, the optical absorption of the prepared samples was measured. Figure 4 shows the optical absorption of the heterojunction CuO–Cu_2_O thin films as well as the control CuO and Cu_2_O thin films with a thickness of around 500 nm. As can be seen, the integration of the Cu_2_O and CuO thin films improved the optical absorption. Indeed, due to the different refractive index of Cu_2_O (2.262) and CuO (2.654), the reflectance loss of the integrated CuO–Cu_2_O was significantly reduced, hence optical absorption was improved according to the Equation (1) [50]: R(%) = (*n*_o_ − *n*_m_)^2^/(*n*_o_ + *n*_m_)^2^ × 100,(1)
where R, *n_m_*, and *n_o_* are reflectance in percentage, the refractive index of the underlying layer, and the refractive index of the top layer, respectively. 

The visible light driven photocatalytic activity of the Cu_2_O, CuO, and CuO–Cu_2_O thin films were investigated by considering the MB degradation. First, the MB degradation rate without photocatalysts under light irradiation and in the presence of a photocatalyst in the dark condition was studied; the results showed no considerable MB degradation rate, which clearly demonstrated that the MB degradation could only be possible in the presence of photocatalysts and visible light irradiation. To reach the adsorption equilibrium, the photocatalysts was placed in the MB solution under the dark condition for 30 min. 

By proceeding with the photocatalytic reaction, the MB concentration gradually dropped due to the degradation reaction. The MB concentration change as a function of the photocatalytic degradation time was plotted and is shown in Figure 5a. In comparison with CuO and Cu_2_O, the MB degradation rate by the CuO–Cu_2_O sample was 0.0241 min^−1^, which was almost twice that of the other samples. The images of the solution after degradation by CuO–Cu_2_O are shown in Figure 5b; as can be seen, the solution color became light. This high photocatalytic activity of the CuO/Cu_2_O sample can be considered as a suitable strategy for the elimination of organic pollutants from water and wastewater treatment systems at a large-scale. 

The superior visible-light driven photocatalytic activity of CuO–Cu_2_O can be ascribed to its improved photo-generated charge carrier separation. Due to the presence of both types of copper oxides in the CuO–Cu_2_O sample, the interface between these two copper oxides acted as a key parameter for the separation of the photo-generated electrons and holes [51,52]. Meanwhile, based on the difference in the energy levels of their CBs and VBs [53,54,55], this led to a higher absorption of visible light. 

The charge collection efficiency of the prepared samples was evaluated by measuring the IPCE characteristics at 0 V versus RHE. Figure 6 illustrates the IPCE characteristics of the heterojunction CuO–Cu_2_O thin films and the control CuO and Cu_2_O thin films with a thickness of 500 nm. As can be seen, the IPCE of the heterojunction CuO–Cu_2_O thin film was greater than that of the CuO and Cu_2_O thin films over a broad range of wavelengths, indicating an improvement in the charge collection of integrated CuO–Cu_2_O thin films when compared to the control CuO and Cu_2_O samples.

Electrochemical impedance spectroscopy (EIS) was used to investigate the kinetics of the interfacial charge transfer of the prepared samples. The Nyquist plot of the prepared CuO–Cu_2_O thin film as well as those of the CuO and Cu_2_O control samples under standard solar illumination at the potential of 0 V vs. RHE is presented in Figure 7. The semi-circle nature of the Nyquist plot at high frequencies is indicative of the charge transfer process and the diameter of the semi-circle is indicative of the interfacial charge transfer resistance between the electrolyte and photocatalyst (*R_ct_*). As seen in Figure 7, the integration of CuO and Cu_2_O significantly reduced the *R_ct_*, indicating an improvement in the charge transfer, a reduction in the recombination rate, and an improvement in photocatalytic activity. The trend of the *R_ct_* values correlated well with the measured IPCE of the prepared samples. 

To further investigate the impact of the integration of Cu_2_O and CuO on the performance of the prepared photocatalyst, linear voltammetry measurements were performed at the pH of the electrolyte of 5.2. Figure 8a,b show the photoelectrochemical (PEC) current density and the photocorrosion stability of the control CuO and Cu_2_O photocatalysts and the CuO–Cu_2_O photocatalyst, respectively. The thickness of the CuO and Cu_2_O control photocatalysts was set at around 500 nm (similar to the thickness of the CuO–Cu_2_O photocatalyst). The PEC current density was measured under dark and standard solar AM 1.5G illumination of 100 mW/cm^2^ (“light on”) conditions. Photocorrosion stability was measured at a potential of 0.25 V vs. RHE. Photocurrent and photocorrosion stability of a photocatalyst is generally determined by the recombination rate and charge collection efficiency. Enhanced photocurrent and photocorrosion stability of the CuO–Cu_2_O photocatalyst when compared to the control CuO and Cu_2_O photocatalysts indicates a reduction in the recombination rate and an enhancement in the separation of generated electron hole pairs. Furthermore, according to the measured PEC characteristics of the deposited CuO, Cu_2_O, and CuO–Cu_2_O thin films on FTO coated substrates, it can be concluded that fabricated electrodes behaved as photocathodes.

The CuO–Cu_2_O photocatalysis mechanism (based on the CB & VB) are graphically presented in Figure 9. Due to the band gap values of Cu_2_O (2.2 eV) and CuO (1.7 eV), both oxides have a high capacity to light absorption in the visible range and can subsequently generate a photo-generated electron-hole pair under visible light irradiation [56,57]. The band edge positions of materials in a heterojunction net strongly affect the photo-generated charge carrier transfer direction [58,59]. As shown in Figure 9, both the CB and VB of Cu_2_O placed below those of CuO; therefore, the photo-generated electrons transfer took place from the Cu_2_O transferred to the CB of CuO under visible light irradiation; whereas the photo-generated holes from CuO transferred to the VB of Cu_2_O [60,61]. Therefore, due to the transfer of the photo-generated electrons and holes in the CuO–Cu_2_O heterojunction net, a significant increase in the life-time of the photogenerated electron-hole pair was observed. O_2_^•^^−^ and OH^•^, as highly oxidative radical species, were generated with the electrons captured by the adsorbed oxygen molecules and holes trapped by the surface hydroxyl, respectively. 

Generally, the degradation of organic pollutants by semiconductors in aqueous environments results from reactive oxygen species (O_2_^•−^, H_2_O_2_, and OH^•^) and photogenerated hole (h^+^) generation during light irradiation. The relevant reactions can be described as follows:CuO–Cu_2_O + *hv* → e^−^ + h^+^(2)
e^−^ + O_2_ → O_2_^•−^(3)
e^−^ + O_2_^•−^ + 2H^+^ → H_2_O_2_(4)
2e^−^ + HO_2_^•^ + H^+^ → OH^•^ + OH^−^(5)
h^+^, H_2_O_2_ , O_2_^•−^, OH^•^ + MB → products (CO_2_, H_2_O, SO_4_^−2^, NO_3_^−^, NH_4_^+^)(6)

To investigate the role of ROSs in MB photocatalytic degradation, t-butanol (t-OH), ammonium oxalate (AO), catalase (CAT), and benzoquinone (BQ) were used as scavengers of the hydroxyl radicals (OH^•^), photo-generated holes (h^+^), H_2_O_2_, and superoxide radicals (O_2_^•−^), respectively [62]. The MB degradation efficiency by CuO–Cu_2_O in the presence of scavengers is shown in Figure 10. As can be seen, an obvious reduction in the photocatalytic activity of CuO–Cu_2_O was observed in the presence of the t-OH, BQ, and OA, which clearly showed the contribution of OH^•^, superoxide, and h^+^ in the photocatalytic degradation process. However, by adding CAT as a scavenger of H_2_O_2_, a light reduction in degradation efficiency was observed; this obviously indicates that H_2_O_2_ did not play an important role in MB degradation by CuO–Cu_2_O.

To evaluate the capacity of CuO–Cu_2_O for large-scale application, the reusability of this sample was tested for five cycles. Figure 11 presents the degradation rate from the five cycle tests. As can be seen, the degradation rate was repeatable. The reduction in photocatalytic degradation rate after five cycles was less than 5%. 

## 4. Conclusions

In conclusion, an efficient visible-light-driven heterojunction photocatalyst of CuO–Cu_2_O thin films for the photocatalytic degradation of methylene blue was fabricated. It was shown that the integration of Cu_2_O and CuO could significantly increase the charge collection and reduce the recombination rate inside the photocatalyst. Furthermore, it was found that integrating CuO and Cu_2_O could improve the optical absorption and facilitate the charge transfer at the interface between the photocatalyst and electrolyte.

## Figures and Tables

**Figure 1 nanomaterials-09-01011-f001:**
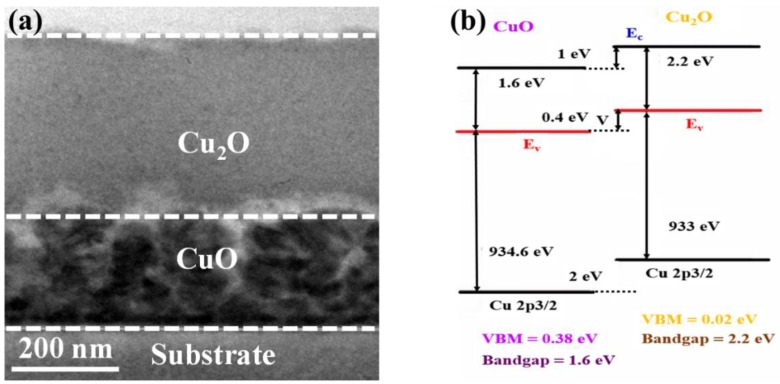
(**a**) Cross-sectional TEM image and (**b**) band alignment of the fabricated heterojunction CuO–Cu_2_O thin film photocatalyst.

**Figure 2 nanomaterials-09-01011-f002:**
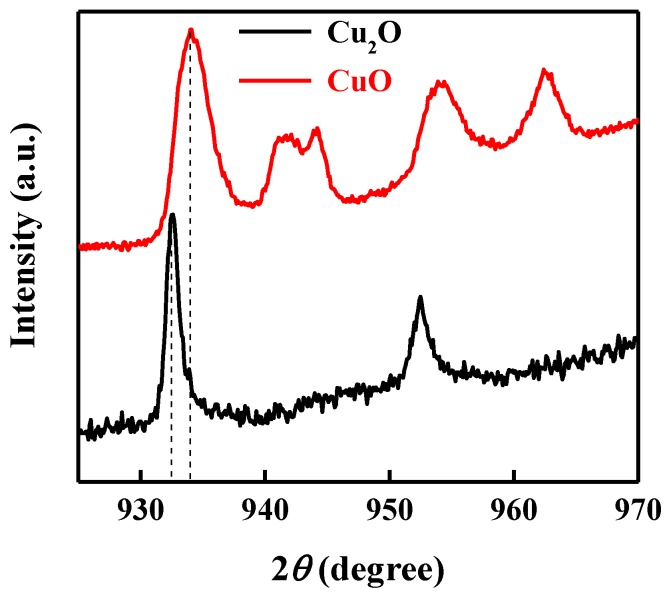
Cu2p XPS spectra of the bottom layer of CuO and the top layer of Cu_2_O.

**Figure 3 nanomaterials-09-01011-f003:**
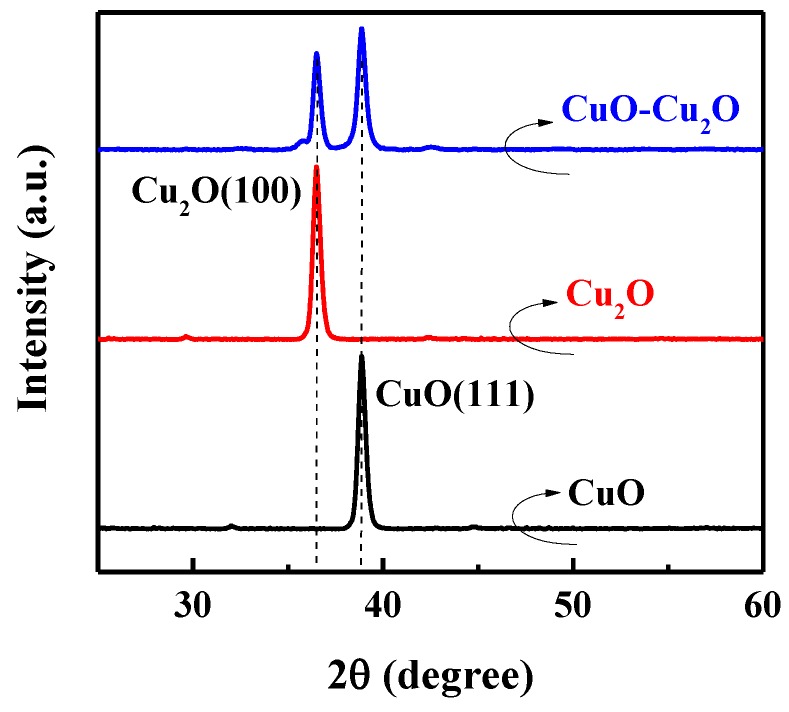
XRD spectra of the CuO–Cu_2_O thin films and control CuO and Cu_2_O thin films with a thickness of 500 nm.

**Figure 4 nanomaterials-09-01011-f004:**
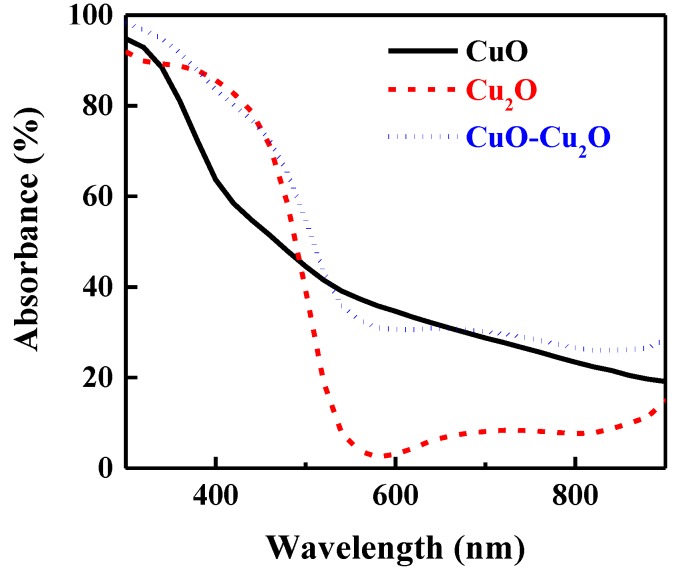
Absorbance spectra of the CuO, Cu_2_O, and the heterojunction CuO–Cu_2_O thin films.

**Figure 5 nanomaterials-09-01011-f005:**
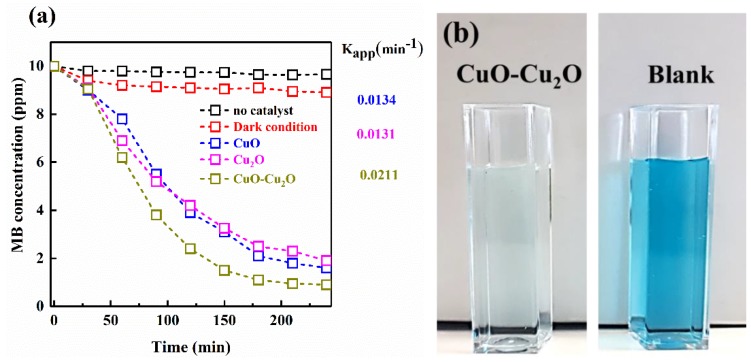
(**a**) Degradation profiles of MB vs. photocatalysis process time. (**b**) Images of the MB solution before and after photocatalysis by CuO–Cu_2_O.

**Figure 6 nanomaterials-09-01011-f006:**
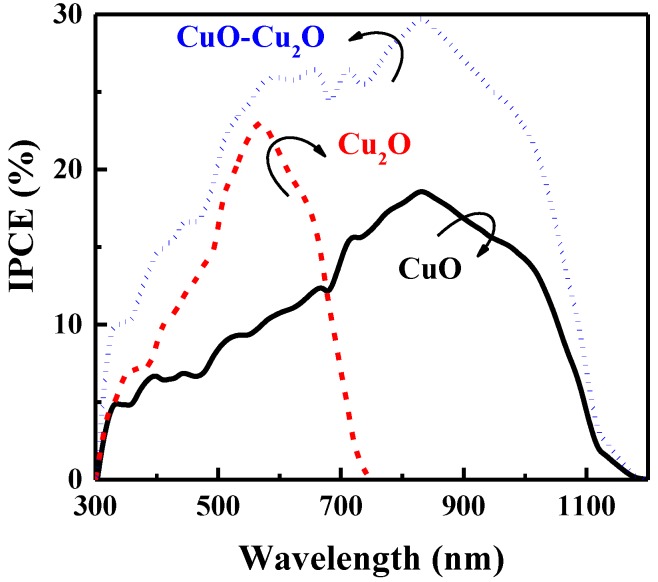
IPCE spectra of the CuO, Cu_2_O, and heterojunction CuO–Cu_2_O thin film photocatalysts.

**Figure 7 nanomaterials-09-01011-f007:**
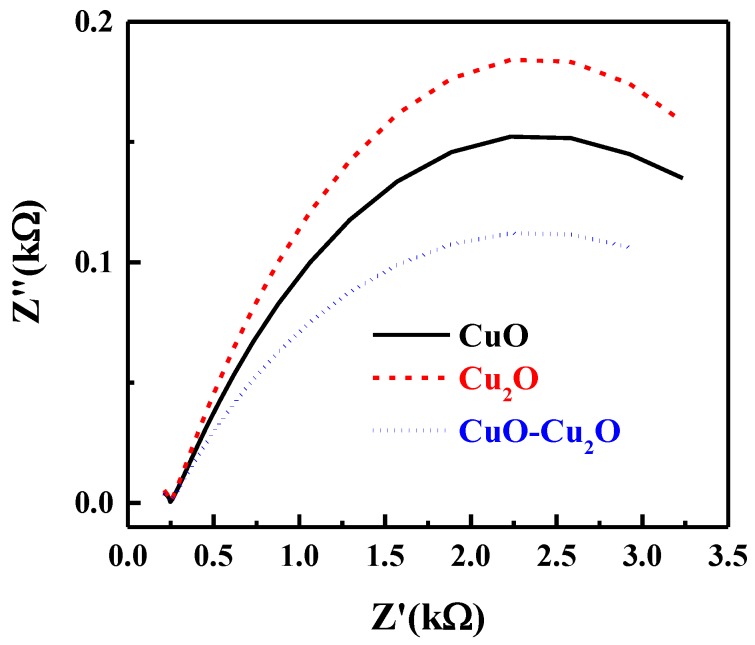
Nyquist plot of the CuO, Cu_2_O and heterojunction CuO–Cu_2_O thin film photocatalysts.

**Figure 8 nanomaterials-09-01011-f008:**
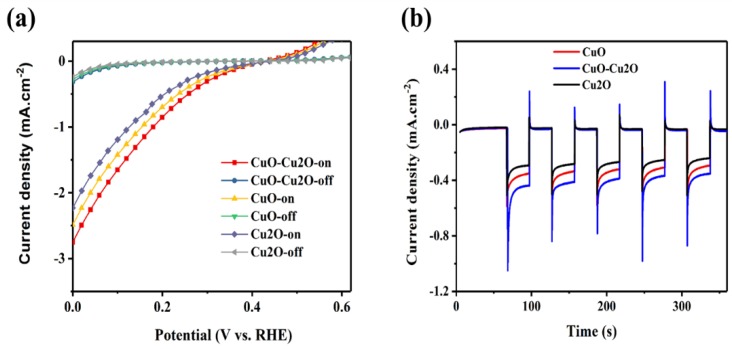
(**a**) photoelectrochemical (PEC) current density and (**b**) photocorrosion stability of the control CuO and Cu_2_O photocatalysts and CuO–Cu_2_O photocatalyst.

**Figure 9 nanomaterials-09-01011-f009:**
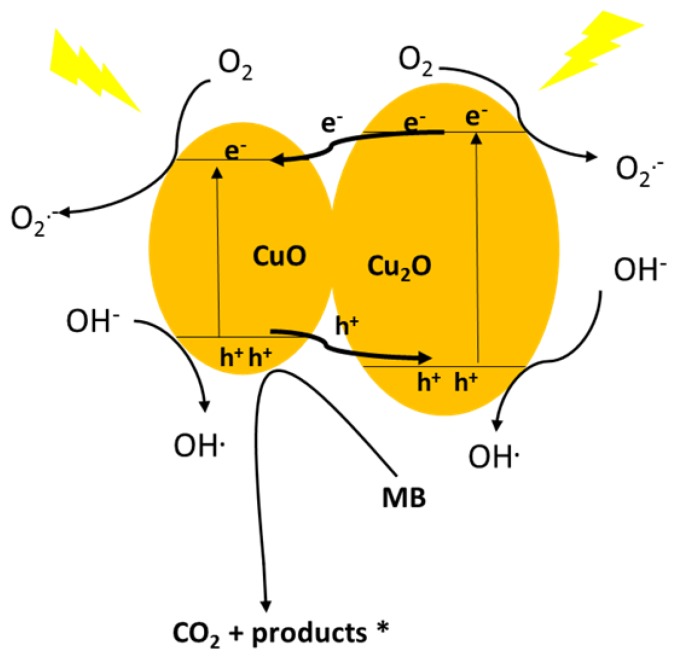
Proposed schematic illustration of the band structure related photocatalytic mechanism for the CuO–Cu_2_O heterojunction net.

**Figure 10 nanomaterials-09-01011-f010:**
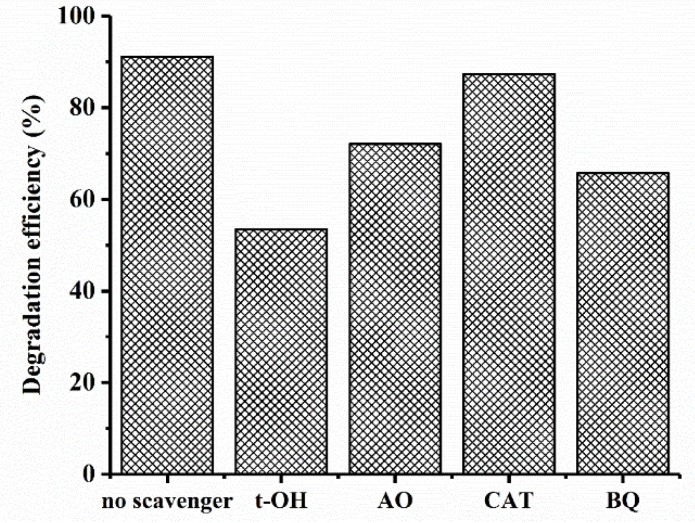
Effect of scavengers on the photocatalytic activity of CuO–Cu_2_O.

**Figure 11 nanomaterials-09-01011-f011:**
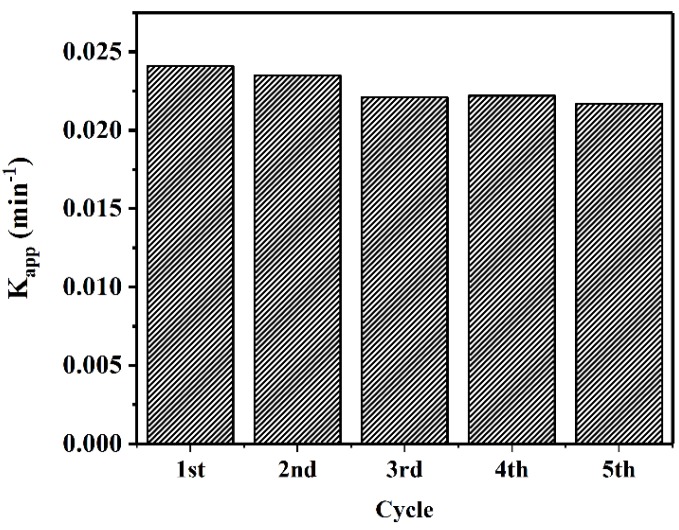
Reusability of the CuO–Cu_2_O sample under visible light irradiation during MB degradation.

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
