# Peer review of "Visible Light Driven Heterojunction Photocatalyst of CuO–Cu_2_O Thin Films for Photocatalytic Degradation of Organic Pollutants"

_nanomaterials, 2019, doi:10.3390/nano9071011_

Round 1
Reviewer 1 Report
This paper is interesting but, requires some extra work to be publishable:
1 - The authors often speak about Copper oxide without specifying whether it is CuO or Cu2O. Since both oxides are used in this article this generates confusion in the reader. There are several paragraphs where this imprecision should be corrected like: (abstract, lines 13-14, introduction line 35 etc ..)
2 - CuO is an n-type semiconductor, while Cu2O is usually p-type. This aspect, very important, should be mentioned by the authors and they should also justify why the have decided to deposit Cu2O on CuO and not the opposite. What happens if the opposite configuration (CuO grown on Cu2O) is used. Linear Voltammetry measurements should be included so that it is more clear how the junction behaves.
3 - The potential used to acquire the IPCE data should be specified and the diagram of figure 8 should include the fermi level alignment in the case of p-type and n-type materials.
Author Response
Response Letter and Summary of Changes
We thank the reviewers for their helpful comments and suggestions to improve the manuscript. Changes were made in the revised manuscript (highlighted and underlined in yellow for easy identification) according to the reviewers’ comments (in blue). The following summarizes our responses to the points raised.
This paper is interesting but, requires some extra work to be publishable:
· We appreciate the positive feedback from the reviewer.
1 - The authors often speak about Copper oxide without specifying whether it is CuO or Cu2O. Since both oxides are used in this article this generates confusion in the reader. There are several paragraphs where this imprecision should be corrected like: (abstract, lines 13-14, introduction line 35 etc ..)
· Thanks for the reviewer’s comment. The comment has been implemented in the manuscript.
2 - CuO is an n-type semiconductor, while Cu2O is usually p-type. This aspect, very important, should be mentioned by the authors and they should also justify why the have decided to deposit Cu2O on CuO and not the opposite. What happens if the opposite configuration (CuO grown on Cu2O) is used. Linear Voltammetry measurements should be included so that it is clear how the junction behaves.
· Thanks for the reviewer’s comment.
To further investigate the impact of integration of Cu2O and CuO on the performance of prepared photocatalyst, linear voltammetry measurement was performed at pH of electrolyte of 5.2. Figure 8a &b shows the photoelectrochemical (PEC) current density and photocorrosion stability of control CuO and Cu2O photocatalyst and CuO- Cu2O photocatalyst respectively. Thickness of CuO and Cu2O control photocatalyst was chosen to be around 500 nm (similar to the thickness of CuO- Cu2O photocatalyst). PEC current density was measured under dark and standard solar AM 1.5G illumination of 100 mW/cm2 (“light on”) conditions. Photocorrosion stability was measured at potential of 0.25 V vs. RHE. Photocurrent and photocorrosion stability of photocatalyst generally is determined by recombination rate and charge collection efficiency. Enhanced photocurrent and photocorrosion stability of CuO- Cu2O photocatalyst as compared to the control CuO and Cu2O photocatalyst indicates the reduction of recombination rate and enhancement of separation of generated electron hole pairs.
Figure 8. (a) photoelectrochemical (PEC) current density and (b) photocorrosion stability of control CuO and Cu2O photocatalyst and CuO- Cu2O photocatalyst
3 - The potential used to acquire the IPCE data should be specified and the diagram of Figure 8 should include the fermi level alignment in the case of p-type and n-type materials.
Thanks for the reviewer’s comment.
Charge collection efficiency of prepared samples was evaluated by measuring the IPCE characteristics at 0 V versus RHE.
Due to lower optical absorption of Cu2O than CuO, greater bandgap of Cu2O (around 2.2 eV) than CuO (around 1.6 eV), and longer carrier diffusion length of Cu2O (around 500 nm) than CuO (around 200 nm), Cu2O was deposited on top of CuO thin films (Figure 1b). Indeed if we deposit CuO on Cu2O, majority of carriers may recombine and cannot reach to the conducting electrode which results in reduction of performance heterojunction CuO- Cu2O photocatalyst.
Figure 1. (a) Cross-sectional TEM image and (b) band alignment of fabricated heterojunction CuO-Cu2O thin film photocatalyst.

Reviewer 2 Report
In this manuscript, vis-light driven photocatalyst of CuO_Cu2O thin film was synthesized and their photocatalytic activity in the reaction of Methylene blue decomposition in comparison to the CuO and Cu2O was studied. I recommended publication of this manuscript after MINOR REVISION. See my suggestions below:
Introduction – it is very generally written. It should be introduced to the subject of studies. The literature on similar photocatalyst materials should be much more extensive. A review of literature reports on copper oxides in photocatalytic processes should be added.
Materials and methods - information about the lamp that was used in photocatalytic reactions should be added.
Figure 5 – Why the photocatalytic reaction in the presence of CuO-Cu2O was stopped after about 170 min? All photocatalytic data should be presented in the same Vis light illumination period of time.
Page 9 - “During the photocatalysis, the ·O2− and ·OH as highly oxidative radical species were generated with the electrons capturing by the adsorbed oxygen molecules and holes trapping by the surface hydroxyl, respectively”. – there is no evidence that O2− and ·OH is main species for MB decomposition. Here, the photoreactions with reactive oxidative species (ROS) should be investigated.
Page 10 – reusability tests - how catalyst was regenerated after each cycle? This should be added to the text.
Author Response
Response Letter and Summary of Changes
We thank the reviewers for their helpful comments and suggestions to improve the manuscript. Changes were made in the revised manuscript (highlighted and underlined in yellow for easy identification) according to the reviewers’ comments (in blue). The following summarizes our responses to the points raised.
Reviewer 2
In this manuscript, vis-light driven photocatalyst of CuO_Cu2O thin film was synthesized and their photocatalytic activity in the reaction of Methylene blue decomposition in comparison to the CuO and Cu2O was studied. I recommended publication of this manuscript after MINOR REVISION. See my suggestions below:
· We appreciate the positive feedback from the reviewer.
Introduction – it is very generally written. It should be introduced to the subject of studies. The literature on similar photocatalyst materials should be much more extensive. A review of literature reports on copper oxides in photocatalytic processes should be added.
· Thanks for the reviewer’s comment.
Several studies reported the application of CuO and Cu2O for photocatalytic degradation of organic pollutants in aqueous media. The CuO/zeoliteX was used for photocatlytic degradation of methylene blue and o-phenylenediamine under sunlight irradiation [42, 43]. CuO/SiO2 was successfully used for MB degradation in a system under UV irradiation and in presence of hydrogen peroxide [44]. In a study, synthesis of the Cu2O/Carbon Nanotubes was reported that indicated considerable photocatalytic activity in phenol degradation in aqueous solution [45]. Katal et al. reported preparation of a high-efficient thin CuO film by in-situ thermal treatment and nanocrystal engineering that indicated superior photocatalytic activity for MB degradation under visible light irradiation [46]. The considerable reusability and stability of this photocatalyst provide a new option for thin film based photocatalyst for industrial applications [46]. In other study, fabrication of the nanocrystalline CuO thin films was performed by using RF magnetron sputtering; based on the some special properties of this photocatalyst including surface defects and oxygen vacancy, CuO thin film indicated remarkable photocatalytic performance for MB degradation [47].
42. Nezamzadeh-Ejhieh, A.; Hushmandrad, S. Solar photodecolorization of methylene blue by CuO/X zeolite as a heterogeneous catalyst. Appl. Catal., A. 2010, 388, 149-159, doi: 10.1016/j.apcata.2010.08.042.
43. Nezamzadeh-Ejhieh, A.; Salimi, Z. Solar photocatalytic degradation of o-phenylenediamine by heterogeneous CuO/X zeolite catalyst. Desalination. 2011, 280, 281–287, doi: 10.1016/j.desal.2011.07.021.
44. Batista, A.P.L.; Carvalho, H.W.P.; Luz, G.H.P. et al. Preparation of CuO/SiO2 and photocatalytic activity by degradation of methylene blue. Environ. Chem. Lett. 2010, 8, 63-67, doi: 10.1007/s10311-008-0192-8
45. Lufeng,Y.; Deqing, C.; Limin, W.; Xu, W.; Junya, L. Synthesis and photocatalytic activity of chrysanthemum-like Cu2O/Carbon Nanotubes nanocomposites. Ceram. Int. 2016, 42, 2502-2509, doi:10.1016/j.ceramint.2015.10.051.
46. Al-Ghamdi, Attieh A.; Khedr, M. H.; Shahnawaze Ansari, M.; Hasan, P. M. Z.; Abdel-wahab, M. Sh.; Farghali, A. A. RF sputtered CuO thin films: Structural, optical and photo-catalytic behavior. Physica. E. Low. Dimens. Syst. Nanostruct. 2016, 81, 83-90, doi: 10.1016/j.physe.2016.03.004.
Materials and methods - information about the lamp that was used in photocatalytic reactions should be added.
· Thanks for the reviewer’s comment.
A 300 W xenon lamp with a cut-off filter was used as visible light source
Figure 5 – Why the photocatalytic reaction in the presence of CuO-Cu2O was stopped after about 170 min? All photocatalytic data should be presented in the same Vis light illumination period of time.
· Thanks for the reviewer’s comment. The comment has been implemented in the manuscript.
Figure 5. (a) Degradation profiles of MB vs photocatalysis process time,
Page 9 - “During the photocatalysis, the ·O2− and ·OH as highly oxidative radical species were generated with the electrons capturing by the adsorbed oxygen molecules and holes trapping by the surface hydroxyl, respectively”. – there is no evidence that O2− and ·OH is main species for MB decomposition. Here, the photoreactions with reactive oxidative species (ROS) should be investigated.
· Thanks for the reviewer’s comment.
Generally, the degradation of organic pollutants by semiconductors in aqueous environments is resulted from reactive oxygen species (O2•−, H2O2, and OH•) and photogenerated hole (h+) generation during light irradiation. The relevant reactions can be described as below:
CuO-Cu2O +hv → e− +h+ (1)
e− +O2 → O2•− (2)
e− +O2•− +2H+ → H2O2 (3)
2e− +HO2• + H+ → OH• + OH− (4)
h+, H2O2,O2•−,OH• +MB → products (CO2, H2O, SO4-2, NO3-, NH4+) (6)
To investigate the role of ROSs in the MB photocatlytic degradation, the t-butanol (t-OH), ammonium oxalate (AO), catalase (CAT), and benzoquinone (BQ) were used as the scavengers of hydroxyl radicals (OH•), photo-generated holes (h+), H2O2, and superoxide radicals (O2•−), respectively [62]. The MB degradation efficiency by CuO-Cu2O in presence of scavengers were shown in Figure 10. As can be seen, an obvious reduction in the photocatalytic activity of CuO-Cu2O was observed in the presence of the t-OH, BQ and OA that clearly showed the contribution of OH•, superoxide and h+ in photocatalytic degradation process. However, by adding CAT as scavenger of the H2O2, a light reduction in degradation efficiency was observed; this obviously indicated that the H2O2 did not play an important role in MB degradation by CuO-Cu2O.
Figure 10. Effect of scavengers on the photocatalytic activity of CuO-Cu2O
62. Gang, X.; Xi, Z.; Wanying, Z.; Shan, Z.; Haijia, S.; Tianwei, Tan. Visible-light-mediated synergistic photocatalytic antimicrobial effects and mechanism of Ag-nanoparticles@chitosan–TiO2 organic–inorganic composites for water disinfection. Appl. Catal. B Environ. 2015, 170–171, 255-262, doi: 10.1016/j.apcatb.2015.01.042.
Page 10 – reusability tests - how catalyst was regenerated after each cycle? This should be added to the text.
· Thanks for the reviewer’s comment.
For the photocatalysts regeneration, any special operation was not carried out; after each experiments, the treated samples was discharged and next run had been performed.

Round 2
Reviewer 1 Report
The authors have answered to all the points raised in my previous review. They should simply specify that according to PEC measurement the electrode behaves as a photocathode.
The paper can be published after minor revisions.
Author Response
We thank the reviewer for their helpful comments and suggestions to improve the manuscript. Changes were made in the revised manuscript (highlighted in green for easy identification) according to the reviewers’ comments (in blue). The following summarizes our responses to the points raised.
The authors have answered all the points raised in my previous review. They should simply specify that according to PEC measurement the electrode behaves as a photocathode.
Thanks for the comment. This comment has been performed in the manuscript.